# The miR-106b/NR2F2-AS1/PLEKHO2 Axis Regulates Migration and Invasion of Colorectal Cancer through the MAPK Pathway

**DOI:** 10.3390/ijms22115877

**Published:** 2021-05-30

**Authors:** Shuzhen Liu, Guoyan An, Qing Cao, Tong Li, Xinyu Jia, Lei Lei

**Affiliations:** 1Key Laboratory of Resource Biology and Biotechnology in Western China, School of Medicine, Northwest University, Ministry of Education, Xi’an 710069, China; LSZ15709483140@163.com (S.L.); 15291032749@163.com (G.A.); caoqing140508028@163.com (Q.C.); litong962021@163.com (T.L.); 2Health Science Center, Xi’an Jiaotong University, Xi’an 710049, China; jb650722@stu.xjtu.edu.cn

**Keywords:** colorectal cancer, migration, NR2F2-AS1, miR-106b, PLEKHO2

## Abstract

Increasing numbers of miRNAs have been observed as oncogenes or tumor suppressors in colorectal cancer (CRC). It was recently reported that hsa-miR-106b-5p (miR-106b) promoted CRC cell migration and invasion. However, there were also studies showing contradictory results. Therefore, in the present study, we further explore the role of miR-106b and its downstream networks in the carcinogenesis of CRC. We observed that the expression of miR-106b is significantly increased in Pan-Cancer and CRC tissues compared with normal tissues from The Cancer Genome Atlas (TCGA) database. Furthermore, we used Transwell, Cell Counting Kit-8, and colony formation assays to clarify that miR-106b promotes the migratory, invasive, and proliferative abilities of CRC cells. For the first time, we systematically screened the target mRNAs and lncRNAs of miR-106b using TCGA database and the bioinformatics algorithms. Dual-luciferase reporter assay confirmed that NR2F2-AS1 and PLEKHO2 are the direct targets of miR-106b. Furthermore, NR2F2-AS1 acts as a competing endogenous RNA (ceRNA) to regulate PLEKHO2 expression by sponging miR-106b. The results of Gene set enrichment analysis (GSEA) and Western blot indicated that they play important roles in CRC progression by regulating MAPK pathway. Thus, miR-106b/NR2F2-AS1/PLEKHO2/MAPK signaling axis may suggest the potential usage in CRC treatment.

## 1. Introduction

Colorectal cancer (CRC) is one of the most common malignant tumors and is the third leading cause of cancer-related deaths worldwide [1]. More and more evidence shows that the carcinogenesis of CRC is related to genetic mutations and epigenetic changes, the latter including non-coding RNA, such as microRNA (miRNAs), long non-coding RNA (lncRNAs) and DNA methylation, etc. [2,3]. MiRNAs are small non-coding RNAs that suppress gene expression by binding to the complementary sequence within the 3-untranslated regions (3′UTR) of target mRNAs [4]. A growing number of miRNAs have been observed as oncogenes or tumor suppressors in various types of cancer, including CRC [5].

It was recently reported that hsa-miR-106b-5p (miR-106b) promoted CRC cell migration and invasion by targeting DLC1 gene [6]. Furthermore, the microarray study indicated that miR-106b was up-regulated in CRC samples [7]. However, previous studies also showed that miR-106b was significantly down-regulated in the stage III-IV of CRC tissues compared with the stage I-II, and miR-106b inhibited the migration and invasion of CRC cells [8]. Therefore, in the present study, we will confirm and further explore the role of miR-106b in the carcinogenesis of CRC.

LncRNAs are defined as transcripts of more than 200 nucleotides without protein-coding functions. Recent researches demonstrated lncRNAs could act as competing endogenous RNAs (ceRNAs) by sharing miRNA response elements [9]. The ceRNA regulatory network of lncRNA–miRNA–mRNA plays an important role in the pathogenesis of cancer [10]. In CRC, lncRNA H19 promotes CRC cell proliferation by competitively binding to miR-200a and derepressing β-catenin [11]. HNF1A-AS1 promotes the metastatic progression of CRC by inhibiting the miR-34a/SIRT1/p53 feedback loop [12].

In the present study, we confirm that miR-106b promotes CRC cell migration and proliferation. We then systematically screened the mRNA and lncRNA targets of miR-106b in CRC using The Cancer Genome Atlas (TCGA) database and bioinformatic analysis. We identified PLEKHO2 as a novel, direct target gene of miR-106b and lncRNA NR2F2-AS1 as a miR-106b sponge using a luciferase reporter assay. In addition, we found that NR2F2-AS1, miR-106b, and PLEKHO2 are involved in the regulation of CRC progression through the MAPK pathway. Our findings may provide new insights into the role of miR-106b in the carcinogenesis of CRC.

## 2. Results

### 2.1. MiR-106b Mediates CRC Cell Migration, Proliferation and Apoptosis

In order to prove that miR-106b is involved in tumorigenesis, we first analyzed miR-106b expression in Pan-Cancer from the TCGA dataset (including 33 types of cancer, 9405 samples), and the result showed that the level of miR-106b in tumor tissues is significantly higher than that in normal tissues (Figure 1A). Then, we found that the expression of miR-106b in CRC tissues is higher than that in colorectal normal tissues from TCGA dataset (including 353 samples) (Figure 1B). We further explored miR-106b expression in 4 CRC cell lines by qRT-PCR. The results showed that miR-106b is highly expressed in HCT-116 and HT-29 cell lines compared with SW480 and LoVo cell lines (Figure 1C). Interestingly, LoVo and HCT-116 cells have higher metastatic abilities compared with the low metastatic cell lines HT-29 and SW480. Therefore, we chose SW480 and HCT-116 cells to transfect miR-106b mimics and inhibitor, respectively (Figure 1D,E). Transwell assays showed that the ability of migration and invasion was increased in SW480 cells transfected with miR-106b mimics, while decreased in HCT-116 cells transfected with miR-106b inhibitor (Figure 1F–I).

To further investigate the proliferative ability of miR-106b, CCK8 and clone formation assays were used. Compared with the negative control, the overexpression of miR-106b promoted SW480 cell proliferation (Figure 2A,C), while transfection of miR-106b inhibitor attenuated HCT-116 cell’s proliferative ability (Figure 2B,D). In addition, the overexpression of miR-106b repressed SW480 cell apoptosis, while the inhibition of miR-106b promoted HCT-116 cell apoptosis by flow cytometry (Figure 2E,F). In summary, these evidences demonstrated that miR-106b promotes CRC progression.

### 2.2. Bioinformatic Analyses Predict the Targets of miR-106b

It was recently reported that lncRNAs could act as ceRNAs by occupying miRNA’s response elements and inhibiting their functions [9]. In this study, we used RNA expression profiles in CRC samples from the TCGA database to investigate differentially expressed mRNAs and lncRNAs. We obtained 8792 down-regulated mRNAs and 764 down-regulated lncRNAs in CRC tissues, and compared with normal tissues. We then systematically screened the mRNAs targeted by miR-106b, and the lncRNAs acted as a miR-106b sponge in CRC using bioinformatic analysis (Figure 3). MiRTarBase, Targetscan and miRDB were used to predict mRNA targets of miR-106b. After we compared the predicted mRNAs in the intersection of three databases with 8792 down-regulated mRNAs, 30 mRNAs were selected for further analysis (Figure 4A). Similarly, we used the DIANA to predict lncRNA targets of miR-106 and chose 18 lncRNAs that were also down-regulated in CRC tissues (Figure 4D). We found that mRNA PLEKHO2 and lncRNA NR2F2-AS1 are among these possible targets of miR-106b. The levels of PLEKHO2 and NR2F2-AS1 are much lower in CRC tissues than in normal tissues (Figure 4B,E). Pearson correlation analysis showed that PLEKHO2 and NR2F2-AS1 were negatively correlated with miR-106b in CRC samples, respectively (Figure 4C,F). In addition, PLEKHO2 was positively correlated with NR2F2-AS1 (Figure 4G). The RT-PCR results also confirmed that the overexpression of miR-106b in SW480 cells inhibited the level of PLEKHO2 and NR2F2-AS1 (Figure 4H,I). These results indicated that NR2F2-AS1 may act as a ceRNA to regulate PLEKHO2 expression by binding miR-106b.

### 2.3. PLEKHO2 Is a Direct Target Gene of miR-106b

To determine whether PLEKHO2 is a direct target gene of miR-106b, we first explored PLEKHO2 expression in CRC cell lines by qRT-PCR (Figure 5A). The results showed that PLEKHO2 mRNA is highly expressed in SW480 and low in HCT-116 cells, which is consistent with protein expression (Appendix A) and is the opposite of miR-106b expression in CRC cells, suggesting miR-106b may inhibit PLEKHO2 expression. We chose SW480 cells to perform the luciferase assay, in which the basal level of miR-106b is lower than that in other cells. The bioinformatic analysis predicted that miR-106b binds to the 3′-UTR of PLEKHO2 (wt-PLEKHO2). We then introduced reverse complementary sequence mutations into the 3′-UTR of PLEKHO2 (mt-PLEKHO2). The luciferase assay showed that miR-106b mimics significantly reduced the luciferase activity of wt-PLEKHO2, but not mt-PLEKHO2 (Figure 5B). These results indicate that PLEKHO2 is the direct target gene of miR-106b.

After we confirmed the suppressive effect of siPLEKHO2-2 in both mRNA and protein level (Figure 5C,D), we further assessed the role of PLEKHO2 in CRC cell migration and invasion using the transwell assay. Compared with the negative control, the inhibition of PLEKHO2 promoted both migration and invasion abilities of HCT-116 cells (Figure 5E,F).

### 2.4. NR2F2-AS1 Is a Direct Target of miR-106b and Regulates PLEKHO2 Expression

To investigate whether NR2F2-AS1 competes with PLEKHO2 for miR-106b binding, we first analyzed NR2F2-AS1 level in 4 CRC cell lines (Figure 6A). Because the level of NR2F2-AS1 in each CRC cell line is similar, SW480 and HCT-116 cell lines were still selected for the luciferase assay and the transwell assay, respectively. The luciferase assay showed that miR-106b mimics significantly reduced the luciferase activity of wt-NR2F2-AS1, but not mt-NR2F2-AS1 (Figure 6B). These findings indicated that NR2F2-AS1 directly interacts with miR-106b.

After we confirmed the inhibitory effect of siNR2F2-AS1-2 using qRT-PCR (Figure 6C), we found that the inhibition of NR2F2-AS1 decreased the protein level of PLEKHO2 (Figure 6D) and promoted both migration and invasion abilities of HCT-116 cells (Figure 6E,F). Furthermore, miR-106b inhibitor significantly increased the mRNA and protein expression of PLEKHO2 in HCT-116 cells (Figure 7A,B). In contrast, the mRNA and protein levels of PLEKHO2 were down-regulated when miR-106b inhibitor was co-transfected with siPLEKHO2 or siNR2F2-AS1 in HCT-116 cells (Figure 7A,B). The transwell assay confirmed that the inhibition of miR-106b reduced migration and invasion abilities of HCT-116 cells. While siPLEKHO2 or siNR2F2-AS1 could rescue the miR-106b inhibitor-induced migration suppression (Figure 7C). All above results suggest that NR2F2-AS1 acts as a ceRNA to regulate PLEKHO2 expression by sponging miR-106b in CRC cells.

### 2.5. MiR-106b/NR2F2-AS1/PLEKHO2 Affect the MAPK Pathway in CRC

In order to figure out the molecular mechanism induced by miR-106b, PLEKHO2 and NR2F2-AS1, we downloaded CRC samples from TCGA database to perform GSEA analysis. The samples were ranked from high to low according to the expression of miR-106b, NR2F2-AS1 and PLEKHO2, respectively. As shown in Figure 8A, the gene set “HALLMARK_MYC_TARGETS_V1” was significantly enriched in the high levels of mR-106b, but enriched in the low levels of NR2F2-AS1 and PLEKHO2. Furthermore, the levels of miR-106b are positively correlated with KRAS and MAP2K1, while NR2F2-AS1 and PLEKHO2 are negatively correlated with KRAS and MYC (Figure 8B). The results of Western blot also confirmed that the suppression of miR-106b weakened the phosphorylation of ERK and the protein expression of MYC, while the inhibitory effect was attenuated by co-expression of miR-106b inhibitor with siPLEKHO2 or siNR2F2-AS1 (Figure 8C). These results suggest that miR-106b, PLEKHO2 and NR2F2-AS1 play important roles in CRC progression by regulating MAPK pathway.

## 3. Discussion

CRC is the third leading cause of cancer-related deaths worldwide [1]. If CRC can be detected at the early stage, the 5-year survival rate can reach 90% or higher [13]. Unfortunately, the early-stage of CRC is often asymptomatic. As a result, more than 60% of CRC is diagnosed at the later stage, and the survival rate of stage IV patients drops to 10% or less [14]. It was recently reported that increasing numbers of miRNAs play important roles in the carcinogenesis and metastasis of CRC [5]. Therefore, studying tumor-related miRNAs and systematically screening their direct targets may provide the possibility for early diagnosis and treatment.

Numerous articles indicated that miR-106b promotes the carcinogenesis of laryngeal cancer, hepatocellular cancer, gastric cancer and CRC [6,15,16,17]. However, some reports revealed that miR-106b inhibits metastasis in breast cancer, endometrial cancer and CRC [8,18,19]. In the present study, we confirmed that miR-106b acts as an oncogene in CRC. In both CRC and Pan-Cancer, miR-106b was significantly up-regulated in tumor tissues compared with normal tissues. We also found that transfection of miR-106b mimics promoted the proliferative and metastatic abilities of SW480 cells, while transfection of miR-106b inhibitor attenuated the proliferative and metastatic abilities of HCT-116 cells. In addition, the overexpression of miR-106b repressed SW480 cell apoptosis, while the inhibition of miR-106b promoted HCT-116 cell apoptosis. These results indicated that miR-106b contributes to CRC progression.

To investigate the underlying mechanism of miR-106b promoting CRC progression, we identified the potential targets of miR-106b. In this study, we not only performed the web-based bioinformatics algorithms, but also systematically screened the target genes from TCGA database. Considering that miR-106b is up-regulated in tumor tissues, the down-regulated mRNAs and lncRNAs in CRC tissues were selected for further analysis. Pearson correlation analysis was performed to ensure that the predicted target mRNAs and lncRNAs were negatively correlated with miR-106b, and positively correlated with each other. Finally, the bioinformatic analysis and qPCR results indicated that NR2F2-AS1 may act as a ceRNA to regulate PLEKHO2 expression by binding miR-106b.

PLEKHO2, a member of the PH-domain-containing protein superfamily, has not been reported to participate in the carcinogenesis of CRC until now. A previous study reported that PLEKHO2 was significantly down-regulated in lung squamous cell carcinoma compared with normal tissues [20]. Furthermore, PLEKHO2 mutant was found in the APC LOH-negative cell lines, indicating that PLEKHO2 may be involved in the tumorigenic activity induced by APC mutations [21]. In addition, increasingly studies showed that PLEKHO1, which shares similar expression profiles with PLEKHO2, suppresses several important pathways, such as PI3K/Akt and TGF-β/BMP signaling pathways [22,23]. And PLEKHO1 acts as a candidate for CRC tumor suppressor [24]. Our results are consistent with these evidences, indicating that PLEKHO2 inhibits both migration and invasion abilities of CRC cells.

LncRNA NR2F2-AS1, located at the chromosome locus 15q26.2, shares a similar tissue-specific expression with NR2F2 [25]. Recently, lncRNA NR2F2-AS1 has been found in the top ten down-regulated lncRNAs in epithelial ovarian cancer, compared with normal ovarian tissues [26]. NR2F2-AS1 was also significantly down-regulated in lung adenocarcinoma tissues both in the TCGA dataset and qRT-PCR experiment [27]. In addition, the results of next-generation sequencing and qPCR validated that NR2F2-AS1 was down-regulated in nasopharyngeal carcinoma [28]. In the present study, we observed that the expression of NR2F2-AS1 was down-regulated in CRC and inversely correlated with the level of miR-106b. The luciferase assay verified that NR2F2-AS1 was a direct target of miR-106b. The suppression of NR2F2-AS1 not only decreased the expression of PLEKHO2, but also promoted migration and invasion abilities of CRC cells. For the first time, these results demonstrate that NR2F2-AS1 regulates PLEKHO2 expression by sponging for miR-106b.

Mutations in KRAS, which lead to hyperactivation of MEK/ERK signaling, occur in 45% of CRC [29]. MAPK signaling promotes the transformation of epithelial characteristics to mesenchymal phenotypes, ultimately leading to tumor migration and invasion [30]. The activation of MAPK pathway could stimulate the transcription factor MYC, which has also been shown to regulate cell proliferation and migration [31,32]. In the present study, our results demonstrated that miR-106b activated the MAPK signaling pathway, while NR2F2-AS1 and PLEKHO2 inhibited the MAPK signaling pathway in CRC. These results were also confirmed by the previous study, which indicated that miR-106b promoted the proliferation and invasion in renal carcinoma cancer through MAPK signaling [33]. Therefore, the miR-106b/NR2F2-AS1/PLEKHO2/MAPK signaling axis may be a novel therapeutic strategy in CRC.

In conclusion, our study revealed that miR-106b significantly promotes CRC cell migration and proliferation. Furthermore, NR2F2-AS1 acts as a ceRNA to regulate PLEKHO2 expression by sponging miR-106b. They played important roles in CRC progression by MAPK pathway. MiR-106b/NR2F2-AS1/PLEKHO2 interaction may suggest the potential usage in CRC treatment.

## 4. Materials and Methods

### 4.1. Bioinformatic Analyses

The expression of miR-106b in Pan-Cancer and CRC was obtained from the UCSC Xena dataset (https://pancanatlas.xenahubs.net, accessed on 28 April 2021). RNA expression profiles (mRNAs and lncRNAs) of CRC samples were downloaded from The Cancer Genome Atlas (TCGA) database, which included 51 normal tissues and 647 CRC tissues. The “edgeR” package in R software was used to analyze the different expression of RNAs (mRNA and lncRNA) (1). Compared to the normal tissue, we obtained 8792 down-regulated mRNAs and 764 down-regulated lncRNAs in CRC tissues, with thresholds of FDR (false discovery rate) < 0.05. miRTarBase (https://bio.tools/mirtarbase, accessed on 28 April 2021), miRDB (http://www.mirdb.org/, accessed on 28 April 2021) and TargetScan (http://www.targetscan.org/vert_72/, accessed on 28 April 2021) were used to predict potential mRNA targets of miR-106b. DIANA (http://carolina.imis.athena-innovation.gr/diana_tools/web/index.php?r=lncbasev2/index, accessed on 28 April 2021) was used to predict potential lncRNA targets of miR-106b. Gene set enrichment analysis (GSEA) was performed to explore the potential mechanism of miR-106b. CRC gene sets were obtained from the TCGA database. We ranked the tumor samples according to the levels of miR-106b, PLEKHO2 or NR2F2-AS1 respectively from high to low. The phenotype label was high-level versus low-level. Data sets and phenotype label files were uploaded to GSEA software (v3.0) for analysis.

### 4.2. Cell Lines and Transfection

Human CRC cell lines SW480, HT-29, LoVo and HCT-116 were purchased from the American Type Culture Collection (ATCC, Rockville, MD, USA) and cultured at 37 °C in a standard incubator containing 5% CO2. MiR-106b mimics, inhibitor, mimics NC: 5′-UUGUACUACACAAAAGUACUG-3′, and inhibitor NC: 5′-CAGUACUUUUGUGUAGUACAA-3′ were synthesized (GenePharma, Shanghai, China) and transfected into CRC cells using the X-tremeGENE siRNA Transfection Reagent (Roche, Mannheim, Germany). To knockdown the expression of PLEKHO2 (ENSG00000241839; GeneID:80301) or NR2F2-AS1 (ENSG00000247809; GeneID:644192), siRNAs were synthesized as follows: siPLEKHO2-1: 5′-GCACCCUGUUCUGAGACUUTT-3′; siPLEKHO2-2: 5′-GGAACAAGGUCAGCGACGUTT-3′; siPLEKHO2-3: 5′-GCGCCUGGAUCUUGAUGUUTT-3′; siNR2F2-AS1-1: 5′-GCAUAGGAGAACGAAACUUTT-3′; siNR2F2-AS1-2: 5′-GGUUCAGCUGCUAACCUUUTT-3′; siNR2F2-AS1-3: 5′-GCACCGUUAUUACUGAAUUTT-3′; siNC: 5′-UUCUCCGAACGUGUCACGUTT-3′.

### 4.3. RNA Extraction and Quantitative RT-PCR

Total RNA of CRC cells was extracted by Trizol reagent (Invitrogen, Carlsbad, CA, USA). The reverse transcription of miRNA was performed using the First Strand microRNA cDNA Synthesis Kit (Sangon, Shanghai, China), which included the forward primer of U6 and the universal reverse primer. The cDNA of mRNA was synthesized using the PrimeScriptTM RT Reagent Kit (Takara, Dalian, China) according to the manufacturer’s instructions. Quantitative real-time PCR (qRT-PCR) was performed using SybrGeen qPCR Mastermix (DBI bioscience, Ludwigshafen, Germany) on CFX96 Touch Real-Time PCR Detection System (Bio-Rad, Hercules, CA, USA). GAPDH and U6 were selected as the internal control. The primers were used in this study as follows: miR106b-F: 5′-GCGTAAAGTGCTGACAGTGCAGAT-3′; PLEKHO2-F: 5′-GAGCTGGGCAGCTATGAGAA-3′; PLEKHO2-R: 5′-GCCTGGAATTTGATGTCGC-3′; NR2F2-AS1-F: 5′-CTCTGGGAATCGTCCTGTATGC-3′; NR2F2-AS1-R: 5′-TGGTTTCCTGGTTCTCTGCC-3′; GAPDH-F: 5′-TGGAAATCCCATCACCATCT-3′; GAPDH-R: 5′-TGGACTCCACGACGTACTCA-3′.

### 4.4. Cell Proliferation and Colony Formation Assay

Cell proliferation ability was evaluated using Cell Counting Kit-8 (CCK8, Dojindo, Kyushu, Japan) according to the manufacturer’s instructions. For the colony formation assay, 1500 transfected cells were incubated in 6-well plates for 2 weeks and then stained with 0.1% crystal violet. The colonies (>50 cells) were imaged and counted. Three independent experiments were performed for each result.

### 4.5. Migration and Invasion Assays

For the migration assays, 8 × 104 SW480 or 5 × 104 HCT116 cells were seeded into the upper of Transwell chambers (Corning, MA, USA). For the invasion assays, the upper chambers were coated with Matrigel (BD Bioscience, Bedford, MA, USA). After 48h, migrated cells were stained and counted under a microscope (Nikon Eclipse TE300, Tokyo, Japan). The independent experiments were repeated three times.

### 4.6. Western Blot Analysis

Total protein was extracted from CRC cells after 48h transfection. The Western blot assay was performed as previously described [34]. Immunoblotting was carried out with anti-PLEKHO2 (1:500, sc-100412, SANTA CRUZ, CA, USA), anti-ERK1/ERK2(1:1000, AF1576, R&D SYSTEMS, MN, USA), anti-pERK (1:500, AF1018, R&D SYSTEMS, MN, USA), anti-alpha Actinin1(1:400, AF8279, R&D SYSTEMS, MN, USA) and anti-c Myc (1:400, sc-40, SANTA CRUZ, CA, USA).

### 4.7. Dual-Luciferase Reporter Assay

The 3′UTR of PLEKHO2 (372bp; ENSG00000241839; GeneID:80301) or NR2F2-AS1 fragment (311bp; ENSG00000247809; GeneID:644192) was inserted into the pmiRGLO Dual-luciferase reporter plasmid (Promega, Madison, WI, USA). The mutant plasmids (mt-PLEKHO2 and mt-NR2F2-AS1) were constructed by the KOD-Plus Mutagenesis Kit (Toyobo, Osaka, Japan). All the plasmids were verified by sequencing. PmiRGLO plasmid (wild type or mutant) was co-transfected with miR-106b mimics or NC using Lipofectamine 2000 (Invitrogen, Carlsbad, CA, USA). After 48h, the luciferase activities were measured using the Dual-Glo Luciferase Assay (Promega, USA) on Luminoskan Ascent (Thermo, Madison, WI, USA), according to the manufacturer’s instructions. The primers were used as follows: wt-PLEKHO2-F: 5′-GTTTAAACGAGCTCGCTAGCGGGGTATGTTGGAATCCGAAGC-3′; wt-PLEKHO2-R: 5′-TTGCATGCCTGCAGGTCGACAGGCCAGGGCATGTTGCTG-3′; mt-PLEKHO2-F: 5′-TGTAAAGTGCTGTTTACTGAAAGAGAGAAAGGGGGGG-3′; mt-PLEKHO2-R: 5′-TGCAGAAATCTGGGCAGGTCC-3′; wt-NR2F2-AS1-F: 5′-GTTTAAACGAGCTCGCTAGCTGTGAATCAGTAAACGTACTAGA-3′; wt-NR2F2-AS1-R: 5′-TTGCATGCCTGCAGGTCGACTAAAAGTGCTGCCCAAGA-3′; mt-NR2F2-AS1-F: 5′-AAAGTGCTGTAGACCTGCAGGCATGACAGCTGAT-3′; mt-NR2F2-AS1-R: 5′-CCCAAGATTGATTGCTCTGATCTG-3′.

### 4.8. Apoptosis Assay

After 72 h of transfection, CRC cells were stained with the Annexin V-FITC/PI apoptosis detection kit (7Sea Pharmatech Co., Ltd.,Shanghai China) according to manufacturer’s instructions. Cell apoptosis was detected by FACSCalibur flow cytometer (BD Bioscience, Bedford, MA, USA). The results of the experiment were analyzed using FlowJo v10 (Tree Star, Ashland, OR, USA).

### 4.9. Statistical Analysis

Data in the figures were shown as the mean ± SD and analyzed by the GraphPad Prism 6 (GraphPad Software Inc. San Diego, CA, USA). The statistical differences were performed by Student’s t-test. The correlation between the expression of miR-106b and PLEKHO2 or NR2F2-AS1 was measured using Pearson’s test. The statistically significance was defined as *p* < 0.05.

## Figures and Tables

**Figure 1 ijms-22-05877-f001:**
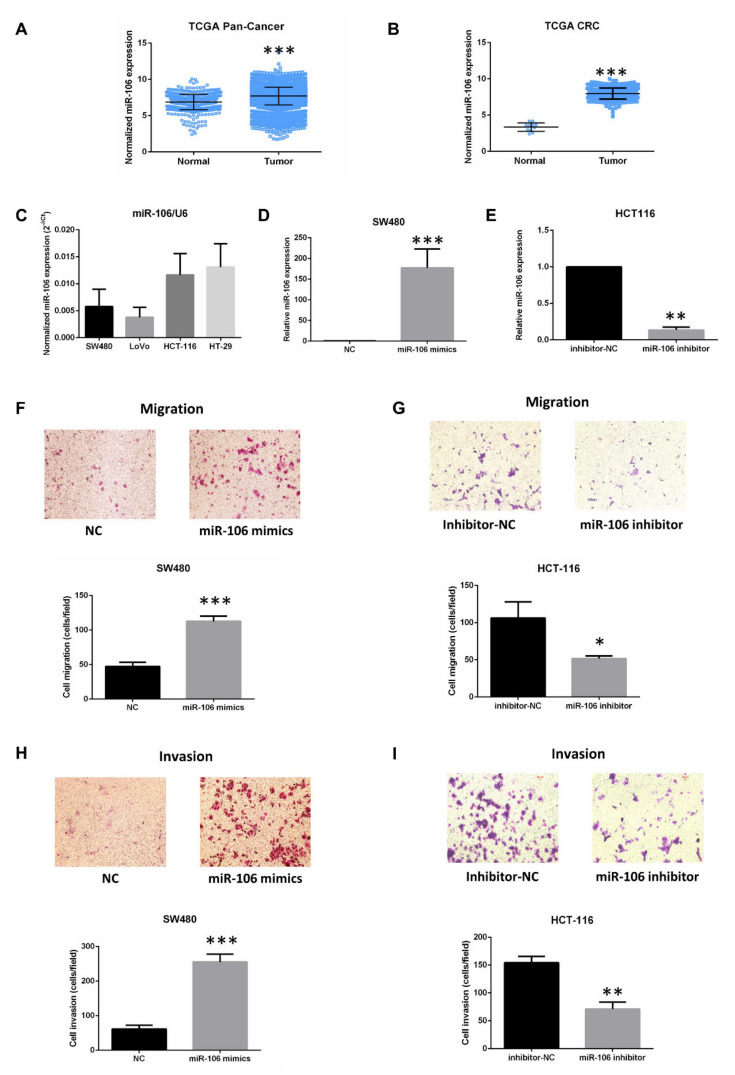
MiR-106b promotes CRC cell migration and invasion. (**A**,**B**) MiR-106b expression analysis in Pan-Cancer and CRC from TCGA dataset. (**C**) The levels of miR-106b in 4 CRC cell lines were analyzed by qPCR. U6 served as an internal control. (**D**) Transfection efficiency of miR-106b mimics in SW480 cells was analyzed by qPCR. (**E**) Transfection efficiency of miR-106b inhibitor in HCT-116 cells was analyzed by qPCR. The data were normalized to U6 and were expressed as fold increase relative to the negative control. (**F**,**H**) Transwell assay showed that the overexpression of miR-106b promotes SW480 cell migration and invasion compared with the negative control. (**G**,**I**) The suppression of miR-106b inhibits HCT-116 cell migration and invasion. All data are shown as the mean ± SD (*n* = 3), * *p* < 0.05, ** *p* < 0.01, *** *p* < 0.001.

**Figure 2 ijms-22-05877-f002:**
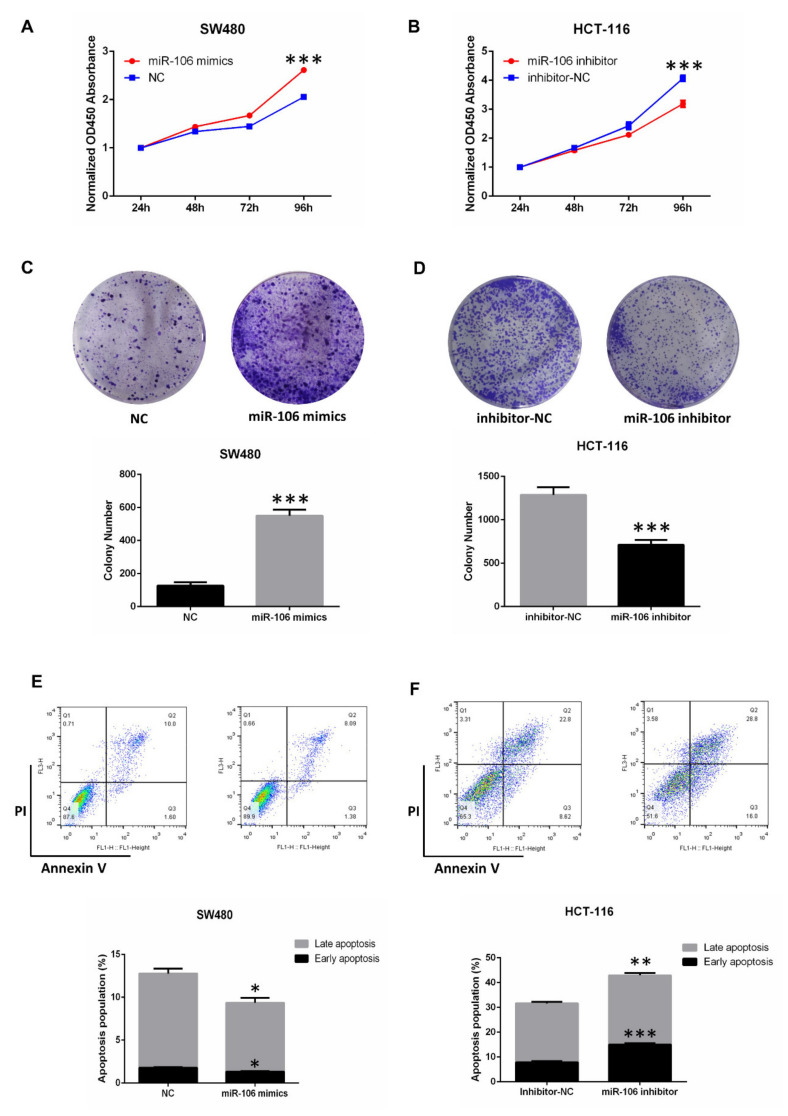
MiR-106b promotes CRC cell proliferation. (**A**,**B**) A CCK8 assay was used to assess the role of miR-106b mimics or inhibitor in CRC cell proliferation. (**C**,**D**) A colony formation assay was used to assess the role of miR-106b mimics or inhibitor in CRC cell proliferation. (**E**) The overexpression of miR-106b inhibits apoptosis in SW480 cells compared with the NC group. (**F**) The inhibition of miR-106b promotes apoptosis in HCT-116 cells. All data are shown as the mean ± SD (*n* = 3), * *p* < 0.05, ** *p* < 0.01, *** *p* < 0.001.

**Figure 3 ijms-22-05877-f003:**
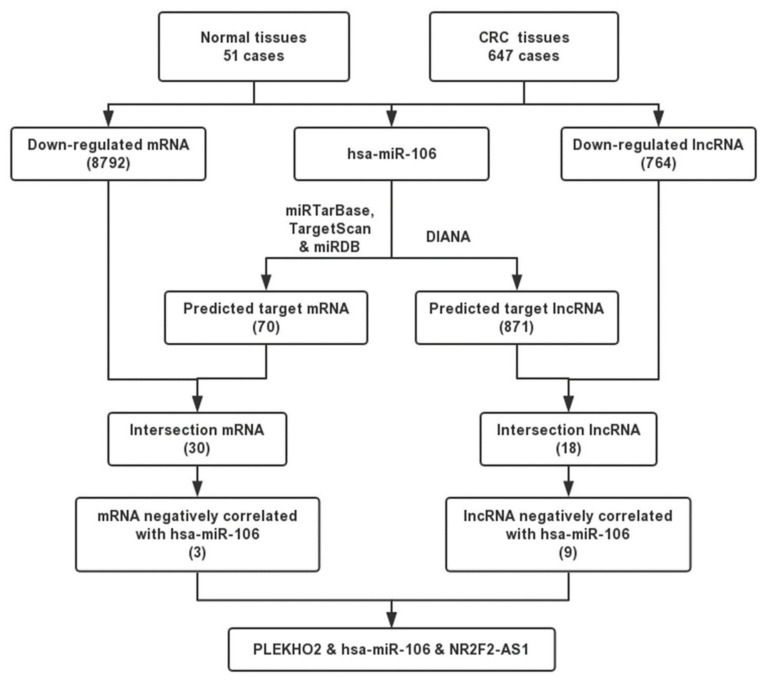
Flow chart of predicting mRNAs and lncRNAs target of miR-106b.

**Figure 4 ijms-22-05877-f004:**
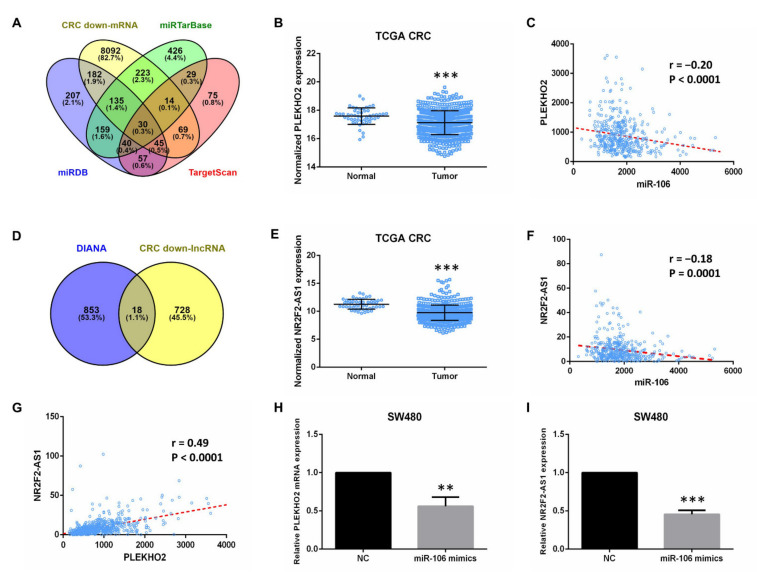
Bioinformatic analyses predict the target of miR-106b. (**A**) The Venn diagram represents the number of miR-106b target mRNAs predicted by miRTarBase, TargerScan, miRDB and down-regulated mRNAs in CRC from TCGA dataset. (**B**) The expression of PLEKHO2 in normal colorectal tissues and CRC from TCGA. (**C**) Pearson’s correlation analysis revealed the expression of PLEKHO2 is negatively correlated with miR-106b in CRC tissues. (**D**) The Venn diagram represents the number of miR-106b target lncRNAs predicted by DIANA and down-regulated lncRNAs in CRC from TCGA dataset. (**E**) The level of NR2F2-AS1 in normal colorectal tissues and CRC from TCGA. (**F**) Pearson’s correlation analysis revealed the level of NR2F2-AS1 is negatively associated with miR-106b in CRC tissues. (**G**) The expression of PLEKHO2 is positively correlated with NR2F2-AS1. (**H**,**I**) Transfection of miR-106b mimics in SW480 cells inhibits the expression of PLEKHO2 and NR2F2-AS1. All data are shown as the mean ± SD, ** *p* < 0.01, *** *p* < 0.001.

**Figure 5 ijms-22-05877-f005:**
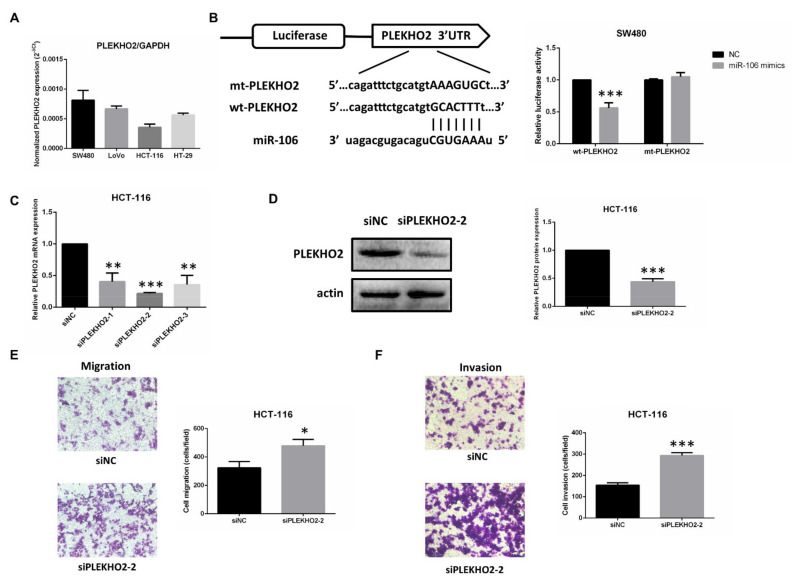
PLEKHO2 is a direct target gene of miR-106b. (**A**) The mRNA level of PLEKHO2 in 4 CRC cell lines was analyzed by qPCR. GAPDH served as an internal control. (**B**) Schematic illustration shows wild type and mutated 3′-UTR of PLEKHO2. A luciferase activity assay was used to identify the miR-106b binding site. (**C**,**D**) Transfection efficiency of PLEKHO2 siRNA in HCT-116 cells was analyzed by qPCR and Western blot assay. (**E**,**F**) Transwell assay showed that the inhibition of PLEKHO2 in HCT-116 cells promotes both migration and invasion compared with the negative control. All data are shown as the mean ± SD (*n* = 3), * *p* < 0.05, ** *p* < 0.01, *** *p* < 0.001.

**Figure 6 ijms-22-05877-f006:**
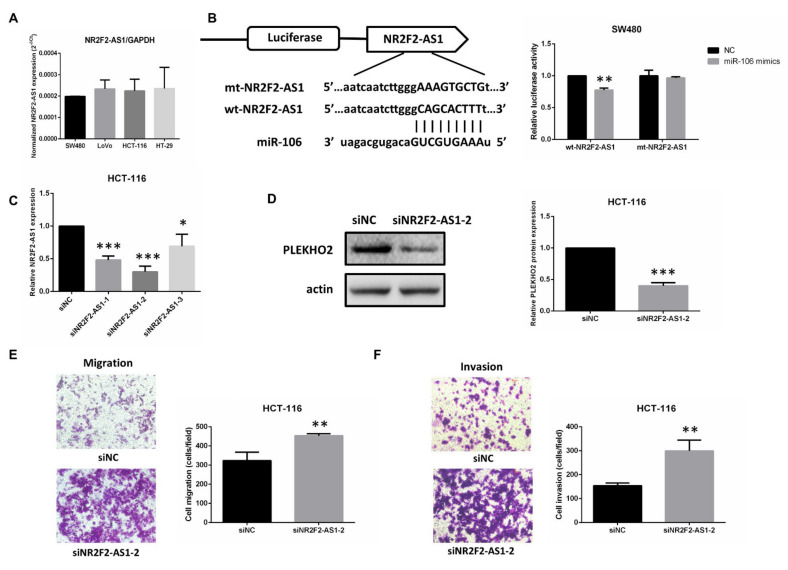
NR2F2-AS1 is a direct target of miR-106b. (**A**) The level of NR2F2-AS1 in 4 CRC cell lines was analyzed by qPCR. GAPDH served as an internal control. (**B**) Schematic illustration shows the miR-106b binding site and the reverse complementary sequence mutations in NR2F2-AS1. A luciferase activity assay was used to identify the miR-106b binding site. (**C**) Transfection efficiency of NR2F2-AS1 siRNA in HCT-116 cells was analyzed by qPCR. (**D**) The protein expression of PLEKHO2 is decreased by transfection of NR2F2-AS1 siRNA. (**E**,**F**) Transwell assay showed that the inhibition of NR2F2-AS1 in HCT-116 cells promotes both migration and invasion compared with the negative control. All data are shown as the mean ± SD (*n* = 3), * *p* < 0.05, ** *p* < 0.01, *** *p* < 0.001.

**Figure 7 ijms-22-05877-f007:**
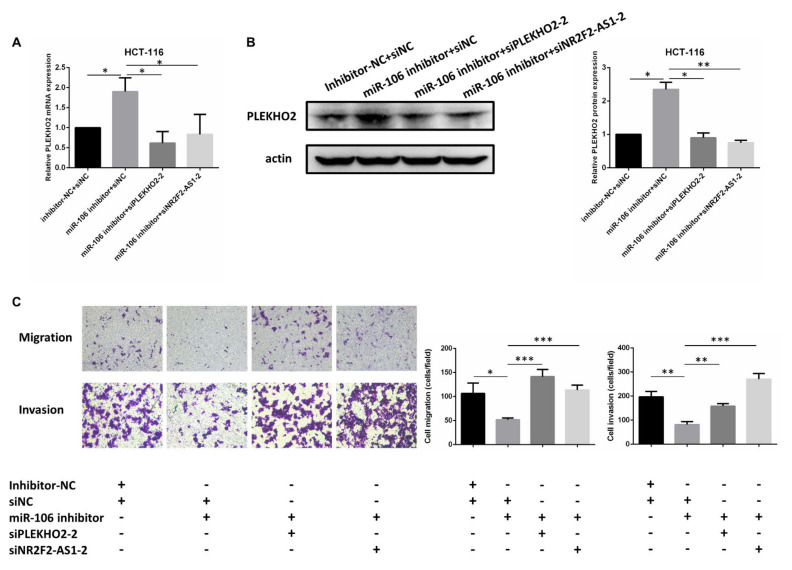
NR2F2-AS1 attenuates PLEKHO2 expression. (**A**,**B**) The mRNA and protein expression of PLEKHO2 was analyzed after co-transfection with PLEKHO2 siRNA or NR2F2-AS1 siRNA and miR-106b inhibitor in HCT-116 cells. (**C**) Transwell assay showed that transfection of miR-106b inhibitor suppressed HCT-116 cell migration and invasion. Co-tranfection of PLEKHO2 siRNA or NR2F2-AS1 siRNA in HCT-116 cells rescues the miR-106b inhibitor-induced migration suppression. All data are shown as the mean ± SD (*n* = 3), * *p* < 0.05, ** *p* < 0.01, *** *p* < 0.001.

**Figure 8 ijms-22-05877-f008:**
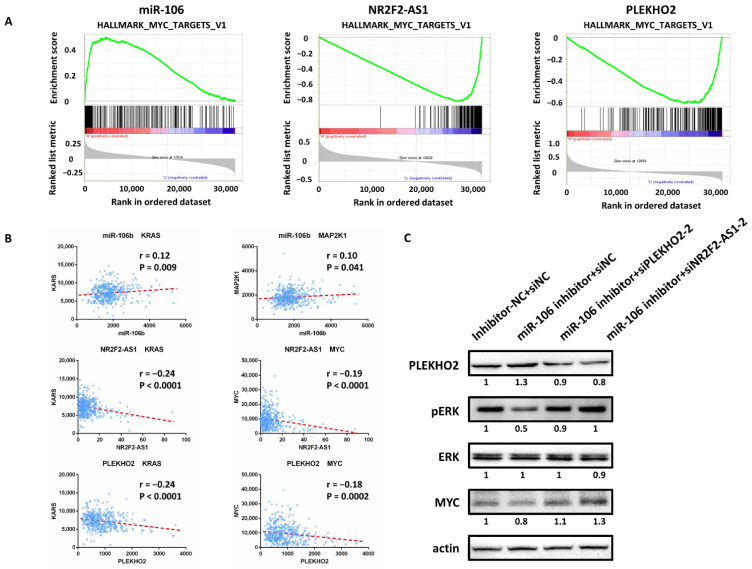
MiR-106b/NR2F2-AS1/PLEKHO2 affect the MAPK pathway in CRC. (**A**) The GSEA results showed a correlation between the levels of MiR-106b/NR2F2-AS1/PLEKHO2 and the MAPK signaling. The gene set “HALLMARK_MYC_TARGETS_V1” is significantly enriched in high levels of mR-106b, but enriched in the low levels of NR2F2-AS1 and PLEKHO2. (**B**) Pearson’s correlation analysis shows that miR-106b is positively correlated with KRAS and MAP2K1. While NR2F2-AS1 and PLEKHO2 are negatively correlated with KRAS and MYC. (**C**) Western blot showed that transfection of miR-106b inhibitor suppresses the phosphorylation of ERK and the protein expression of MYC. Co-tranfection of PLEKHO2 siRNA or NR2F2-AS1 siRNA rescues miR-106b inhibitor-induced expression suppression.

## Data Availability

Not applicable.

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
