# Peer review of "The miR-106b/NR2F2-AS1/PLEKHO2 Axis Regulates Migration and Invasion of Colorectal Cancer through the MAPK Pathway"

_ijms, 2021, doi:10.3390/ijms22115877_

Round 1
Reviewer 1 Report
The authors provide exciting work on a miRNA ceRNA axis in colon cancer. Their work shows that the interaction between mir-106b and PLekho2-NR2F2-AS1 is essential for cell invasion. The work is extensive and worthy of publication. Some comments that should be addressed are below.
- the authors can include the proper names of miRNA as has-mir-106b-5p initially and then shorten it. Please the miRbase accession and genebank/Ensembl accession of plekho2 and NR2F2, which were used to design the miRNA binding sites.
- How was the TCGA data analyzed. Please provide a bookmark to Xenabrowser or similar and mention the specific dataset used for the analysis. Please also provide how downregulated lncRNA/mRNA list was generated in the method.
- for figure 2, the p-value is for late or early apoptosis or both? For all figures, please mention all statistics performed on the data.
- The authors can show the protein levels of plexkho2 in SW840, since hct1116 has a good amount of pleckho2 despite being high in mir-106b (figure 5d).
- Figure 7B: it is hard to see the difference between the last two columns. If mir016 inhibitor is used, the n siNR2F2 should not have an effect and the levels should be similar to 2nd column mir-106 inhibitor only. How do the authors explain this?
- figure 7C: similar issued as point 5 above. siNR2F2 should not increase invasion when in combination with mir-106b inhibitors. Is NR2F2 affecting other pathways by sponging other miRNAs?
- Figure 8A, did the authors see any effect on the MAPK pathway in GSEA for eg. “KEGG_MAPK_SIGNALING_PATHWAY”. Was the GSEA plot generated with normal-tumor sample comparison? How is high to low expression determined for the genes?
- Figure 8C, the reduction In pERK is difficult to see in the figure. Can they provide better figures and possibly quantify the bands? How does pERK change with siPLEKH2O2 only?
Author Response
Response to Reviewer 1 Comments
The authors provide exciting work on a miRNA ceRNA axis in colon cancer. Their work shows that the interaction between mir-106b and PLekho2-NR2F2-AS1 is essential for cell invasion. The work is extensive and worthy of publication. Some comments that should be addressed are below.
- the authors can include the proper names of miRNA as has-mir-106b-5p initially and then shorten it. Please the miRbase accession and genebank/Ensembl accession of plekho2 and NR2F2, which were used to design the miRNA binding sites.
Answer: We apologize for the poor description of the proper name of miR-106b and the accession of genes. We have added the information in the revised manuscript. “It was recently reported that hsa-miR-106b-5p (miR-106b) promoted CRC cell migration and invasion by targeting DLC1 gene.”“The 3’UTR of PLEKHO2 (372bp; ENSG00000241839; GeneID:80301) or NR2F2-AS1 fragment (311bp; ENSG00000247809; GeneID:644192) was inserted into the pmiRGLO Dual-luciferase reporter plasmid (Promega, USA).”
- How was the TCGA data analyzed. Please provide a bookmark to Xenabrowser or similar and mention the specific dataset used for the analysis. Please also provide how downregulated lncRNA/mRNA list was generated in the method.
Answer: We apologize for the poor description of the Methods. We have added the information in the Methods in the revised manuscript.
“The expression of miR-106b in Pan-Cancer and CRC was obtained from the UCSC Xena dataset (https://xenabrowser.net/datapages/?cohort=TCGA%20Pan-Cancer%20(PANCAN)&removeHub=https%3A%2F%2Fxena.treehouse.gi.ucsc.edu%3A443).RNA expression profiles (mRNAs and lncRNAs) of CRC samples were downloaded from The Cancer Genome Atlas (TCGA) database, which included 51 normal tissues and 647 CRC tissues. The “edgeR” package in R software was used to analyze the different expression of RNAs (mRNA and lncRNA) (1). Compared to the normal tissue, we obtained 8792 down-regulated mRNAs and 764 down-regulated lncRNAs in CRC tissues, with thresholds of FDR (false discovery rate) < 0.05. miRTarBase (https://bio.tools/mirtarbase), miRDB (http://www.mirdb.org/) and TargetScan (http://www.targetscan.org/vert_72/) were used to predict potential mRNA targets of miR-106b. DIANA (http://carolina.imis.athena-innovation.gr/diana_tools/web/index.php?r=lncbasev2/index) was used to predict potential lncRNA targets of miR-106b.”
- for figure 2, the p-value is for late or early apoptosis or both? For all figures, please mention all statistics performed on the data.
Answer: We apologize for the confusion caused by the lack of statistics in figure 2. We have added all statistics in the revised figure.
- The authors can show the protein levels of plexkho2 in SW840, since hct1116 has a good amount of pleckho2 despite being high in mir-106b (figure 5d).
Answer: As suggested by reviewer, we have analyzed the protein levels of PLEKHO2 in both SW480 and HCT-116 cell lines and added this figure in supplementary figure 1. As shown in figure, the expression of PLEKHO2 is a little higher in SW480 cells than in HCT-116 cells, which is consistent with the qPCR results. And there are amount of PLEKHO2 in HCT-116 cells, which allows us to perform the following experiments.
- Figure 7B: it is hard to see the difference between the last two columns. If mir016 inhibitor is used, the n siNR2F2 should not have an effect and the levels should be similar to 2ndcolumn mir-106 inhibitor only. How do the authors explain this?
Answer: We would like to thank the reviewer for this question. It was reported that some miRNA inhibitors can sequester the miRNA without causing degradation, thus measuring inhibition of miRNA activity represents a significant challenge (2). In our article, although miR-106 inhibitor is effective (figure 1 E, G, I and figure 2 B, D, F), we can’t confirm that all miR-106 in the cell has been sequestered or degraded by the inhibitor. Therefore, NR2F2-AS1 might still bind to some miR-106. Co-transfection of siNR2F2-AS1 could release some miR-106, which may attenuate the inhibitory effect by miR-106 inhibitor.
We also find the article showing the similar results (3), in which compared with miR-34a inhibitor, co-transfecting siSNHG7 (lncRNA) andmiR-34a inhibitor downregulated the mRNA expression of GALNT7 (figure 6e shown in coverletter).
- figure 7C: similar issued as point 5 above. siNR2F2 should not increase invasion when in combination with mir-106b inhibitors. Is NR2F2 affecting other pathways by sponging other miRNAs?
Answer: Please also see our response in question 5. If not all miR-106 in the cell has been sequestered or degraded by the inhibitor, NR2F2-AS1 might still bind to some miR-106. Co-transfection of siNR2F2-AS1 could release some miR-106, which could increase the migration and invasion of CRC cells. NR2F2-AS1 sponging other miRNAs might be another possible reason for increased invasion.
There is also the article (3) showing that compared with miR-34a inhibitor, co-transfecting siSNHG7 (lncRNA) andmiR-34a inhibitor decreased the clone number, migration and invaded cell number, average tube length (figure shown in coverletter).
- Figure 8A, did the authors see any effect on the MAPK pathway in GSEA for eg. “KEGG_MAPK_SIGNALING_PATHWAY”. Was the GSEA plot generated with normal-tumor sample comparison? How is high to low expression determined for the genes?
Answer: As suggested by reviewer, we have analyzed the MAPK pathway in GSEA and found that the gene set “HALLMARK_KRAS_SIGNALING_DN” was significantly enriched in the high levels of PLEKHO2 (p<0.001), which means PLEKHO2 inhibits the KRAS signaling pathway. This finding is consistent with our previous results and is added in supplementary figure 1 (figure was also shown in coverletter).
We apologize for the poor description of the GSEA. We have added the information in the Methods in the revised manuscript.
“Gene set enrichment analysis (GSEA) was performed to explore the potential mechanism of miR-106b. CRC gene sets were obtained from the TCGA database. We ranked the tumor samples according to the levels of miR-106b, PLEKHO2 or NR2F2-AS1 respectively from high to low. The phenotype label was high-level versus low-level. Data sets and phenotype label files were uploaded to GSEA software (v3.0) for analysis.”
- Figure 8C, the reduction In pERK is difficult to see in the figure. Can they provide better figures and possibly quantify the bands? How does pERK change with siPLEKH2O2 only?
Answer: We appreciate this suggestion and have repeated the experiment. The better images and quantify numbers were shown in the revised figures. We also analyzed the pERK change with siPLEKHO2 only and added this figure in supplementary figure 1. As shown in figure, transfection of siPLEKHO2 downregulated the expression of pERK compared with siNC, which is consistent with our previous results (figure was also shown in coverletter).
References
- Robinson MD, McCarthy DJ, Smyth GK. edgeR: a Bioconductor package for differential expression analysis of digital gene expression data. Bioinformatics (2010) 26(1):139-40.
- Esau CC. Inhibition of microRNA with antisense oligonucleotides. Methods (2008) 44(1):55-60.
- Li Y, Zeng C, Hu J, Pan Y, Shan Y, Liu B, et al. Long non-coding RNA-SNHG7 acts as a target of miR-34a to increase GALNT7 level and regulate PI3K/Akt/mTOR pathway in colorectal cancer progression. Journal of hematology & oncology (2018) 11(1):1-17.

Reviewer 2 Report
Liu and colleagues presented an interesting work demonstrating a miR-106b/NR2F2-AS1/PLEKHO2 axis which can regulate migration and invasion of colorectal cancer through the MAPK pathway. In general, this work can represent an advance in the knowledge about new molecular/epigenetic mechanisms involved in CRC carcinogenesis. The authors also clarify the role of miR-16 in CRC progression by overcoming the contradictory results available so far.
The manuscript is very well written and theorically well designed, however I have the following major concerns regarding this work.
Point 1
Overall, the description of the key findings of the study and their relevance to a potential clinical application should be thoroughly revised both in the abstract and in the main text. For example, the last sentence of the abstract deserves a revision (line 23-24 “miR-106b/NR2F2-AS1/PLEKHO2/MAPK signaling axis may represent a novel strategy for prognostic prediction and treatment of CRC patients”); the following sentence at line 236 should be scaled down “These results indicated that miR-106b promotes CRC progression”, etc.
Point 2
It would be useful to add a brief description of the four CRC cell lines analyzed in the study to justify and discuss the reason for the main differences between them highlighted by the results.
Point 3
What is the negative control used to obtain the results shown in figures 1, 2, 5 and 6? Please add this definition in material and methods section.
Point 4
In qPCR experiments, a minimum of two housekeeping genes is recommended to make the analysis more robust.
Point 5
At line 234, the authors improperly talk about “metastasis of HCT-116 cells”, please consider to change in “metastatic behavior of..”.
Author Response
Response to Reviewer 2 Comments
Liu and colleagues presented an interesting work demonstrating a miR-106b/NR2F2-AS1/PLEKHO2 axis which can regulate migration and invasion of colorectal cancer through the MAPK pathway. In general, this work can represent an advance in the knowledge about new molecular/epigenetic mechanisms involved in CRC carcinogenesis. The authors also clarify the role of miR-16 in CRC progression by overcoming the contradictory results available so far.
The manuscript is very well written and theorically well designed, however I have the following major concerns regarding this work.
- Overall, the description of the key findings of the study and their relevance to a potential clinical application should be thoroughly revised both in the abstract and in the main text. For example, the last sentence of the abstract deserves a revision (line 23-24 “miR-106b/NR2F2-AS1/PLEKHO2/MAPK signaling axis may represent a novel strategy for prognostic prediction and treatment of CRC patients”); the following sentence at line 236 should be scaled down “These results indicated that miR-106b promotes CRC progression”, etc.
Answer: As suggested by reviewer, we have revised the abstract and the main text of our manuscript. The last sentence in the abstract was changed to “Thus, miR-106b/NR2F2-AS1/PLEKHO2/MAPK signaling axis may suggest the potential usage in CRC treatment.” The sentence at line 236 was changed to “These results indicated that miR-106b contributes to CRC progression.” The sentence at line 290 was changed to “MiR-106b/NR2F2-AS1/PLEKHO2 interaction may suggest the potential usage in CRC treatment.”
- It would be useful to add a brief description of the four CRC cell lines analyzed in the study to justify and discuss the reason for the main differences between them highlighted by the results.
Answer:We apologize for the poor description of the four CRC cell lines. We have added a brief description in the revised manuscript. “Interestingly, LoVo and HCT-116 cells have higher metastatic abilities compared with the low metastatic cell lines HT-29 and SW480.” Therefore, the metastatic ability of cell lines may not be the main reason for the different expression of miR-106b. Furthermore, we chose the low metastatic cell line SW480 to transfect miR-106b mimics and the high metastatic cell line HCT-116 to transfect miR-106b inhibitor to investigate the effect of miR-106b on cell biological behaviors.
- What is the negative control used to obtain the results shown in figures 1, 2, 5 and 6? Please add this definition in material and methods section.
Answer: We apologize for the poor description of the negative control. We have added the information in the material and methods section in the revised manuscript. “mimics NC: 5’-UUGUACUACACAAAAGUACUG-3’, and inhibitor NC: 5’-CAGUACUUUUGUGUAGUACAA-3’. siNC: 5’-UUCUCCGAACGUGUCACGUTT-3’.”
- In qPCR experiments, a minimum of two housekeeping genes is recommended to make the analysis more robust.
Answer: We appreciate the careful review and agree with this comment. At the beginning, we also chose Peptidyl-prolyl isomerase/cyclophilin A (PPIA) as a reference gene for normalization, according to the article (1). When the results showed the similar trend, we only used GAPDH as the reference gene for other qPCR experiments (Figure was shown in coverletter). Thanks for the suggestion. In the future, we will use two reference genes to make the results more reliable.
- At line 234, the authors improperly talk about “metastasis of HCT-116 cells”, please consider to change in “metastatic behavior of..”.
Answer: We appreciate the careful review and suggestions. We have changed “metastasis of HCT-116 cells” for “the proliferative and metastatic abilities of HCT-116 cells”.
References
- Sørby LA, Andersen SN, Bukholm IR, Jacobsen MB. Evaluation of suitable reference genes for normalization of real-time reverse transcription PCR analysis in colon cancer. Journal of Experimental & Clinical Cancer Research (2010) 29(1):1-9.

Round 2
Reviewer 2 Report
The authors have satisfactorily revised the manuscript and all my concerns have been addressed.